# Gender Differences in the Impact of COVID-19 Pandemic on Mental Health of Italian Academic Workers

**DOI:** 10.3390/jpm12040613

**Published:** 2022-04-11

**Authors:** Valentina Giudice, Teresa Iannaccone, Filomena Faiella, Filomena Ferrara, Giusi Aversano, Silvia Coppola, Elisa De Chiara, Maria Grazia Romano, Valeria Conti, Amelia Filippelli

**Affiliations:** 1Department of Medicine, Surgery and Dentistry “Scuola Medica Salernitana”, University of Salerno, 84081 Baronissi, Italy; tiannaccone@unisa.it (T.I.); vconti@unisa.it (V.C.); afilippelli@unisa.it (A.F.); 2Clinical Pharmacology, University Hospital “San Giovanni di Dio e Ruggi d’Aragona”, 84131 Salerno, Italy; 3Department of Humanistic Studies, University of Salerno, 84084 Fisciano, Italy; ffaiella@unisa.it; 4Single Guarantee Committee, University of Salerno, 84084 Fisciano, Italy; fferrara@unisa.it (F.F.); gaversano@unisa.it (G.A.); scoppola@unisa.it (S.C.); eldechia@unisa.it (E.D.C.); maromano@unisa.it (M.G.R.); 5Department of Economy and Statistics, University of Salerno, 84084 Fisciano, Italy

**Keywords:** COVID-19, mental health, gender

## Abstract

The 2020 pandemic for coronavirus SARS-CoV-2 infection has required strict measures for virus spreading reduction, including stay-at-home orders. To explore gender differences in mental health status after the first wave of the pandemic and in teleworking, we analyzed the frequency and distribution of emotions and coping strategies for facing the pandemic stratified by gender using data from an online survey conducted at the University of Salerno, Italy, between 11 May and 10 June 2020. The online questionnaire included 31 items on demographics, teleworking, COVID-19 emergency, and gender-based violence, with multiple-choice answers for some questions. Females felt significantly sadder (*p* = 0.0019), lonelier (*p* = 0.0058), more fearful (*p* = 0.0003), and more insecure (*p* = 0.0129) than males, experienced more sleep disorders (*p* = 0.0030), and were more likely to sanitize surfaces compared to males (*p* < 0.0001). Our results show gender differences in awareness and concerns about the COVID-19 pandemic that differently influenced mood, as females were more frightened and worried than males.

## 1. Introduction

The coronavirus SARS-CoV-2 is responsible for the 2020 pandemic disease known as COVID-19 characterized by severe respiratory symptoms and an acute respiratory distress syndrome (ARDS) requiring intensive care treatment in more than 3% of cases [1,2,3]. COVID-19 incidence does not greatly vary between sexes (48% in males vs. 52% in females); however, males have an increased mortality rate for COVID-19 complications, with a male-to-female fatality ratio of up to 3.5, except for some countries, such as India [4,5,6]. Gender differences in clinical manifestations and outcomes are related to several sex-related features, such as gender-associated angiotensin-converting enzyme 2 (ACE-2) expression, or the influence of sex hormones on immune response and coagulation cascade regulation [7,8,9,10,11]. Cardiovascular and respiratory diseases are well-known risk factors of severe COVID-19, are related to increased mortality and have increased incidence in males compared to females [8]. 

Sex significantly and variously impacts COVID-19 biology and clinical outcomes, and differences are attributable to sex steroid action or other gender-related physiological differences, such as in the type of immune responses [11]. Moreover, variations in females are not only related to sex and age, but also to pregnancy, and pre- or post-menopausal conditions. For example, decreased sex hormone levels are related to severe symptoms and increased mortality rate, as described in older males with low free and total testosterone, or postmenopausal females with low 17-β estradiol [11]. Conversely, immune system assets in females, while favoring a better response against viral agents, can cause an exaggerated response after vaccination, thus leading to increased frequencies of vaccine-related adverse events and low efficacy [11]. However, sex differences on COVID-19 impact are not only related to a different gender-related biology, but also to environment, behavior, type of job, and habits that could also impact COVID-19 incidence and the development of psychological disturbances, such as anxiety, distress, depression, and post-traumatic stress disorder (PSTD) [12,13,14]. Young adults are more vulnerable to long-term COVID-19 sequalae, because they might be more affected by restrictions, social and physical distancing, and/or quarantine, as highlighted by a worldwide increase in suicides among adolescents and young adults during the COVID-19 pandemic [15,16]. Precariousness in financial and professional life could be an additional risk factor for mental health disorders [17]. Among young adults, women have a higher burden compared to men as they also take care of sick loved ones, family, children’s school duties, and/or housekeeping [12]. 

In this study, we investigated gender differences in perceptions and coping strategies during pandemic using data from an online survey conducted after the first pandemic wave (between 11 May through 10 June 2020) on academic staff and students, and in the mental health impact of COVID-19 and the benefits and disadvantages of teleworking during the pandemic. We studied the frequency and distribution of emotions experienced by participants during the stay-at-home mandate, and we considered those feelings as mental health status indicators of COVID-19 impact on the mental health of the general population. Our results stratified by gender show significant different mental health outcomes after the pandemic.

## 2. Materials and Methods

### 2.1. Participants and Survey

A total of 899 participants took part in an anonymous online survey (Appendix A) that was conducted after the first wave of the COVID-19 pandemic, from 11 May to 10 June 2020. The questionnaire was available in Italian and English. Participants were voluntarily recruited among all employers and students at the University of Salerno, Fisciano, Italy, reached by email without any compensation. Demographic characteristics are summarized in Table 1. The survey comprised 31 questions designed based on the Coronavirus Health Impact Survey [18] and included demographics, teleworking, COVID-19 emergency, and domestic violence (Appendix A). On question 22, participants were asked to indicate predominant emotions during pandemic among the following possible choices: resignation; insomnia; loneliness; sadness; exasperation; fear; insecurity; anxiety; concern; tranquility; ease; and pleasure. Participants could choose more than one answer. These feelings were included in the survey and used as mental health status indicators during COVID-19 pandemic based on previous studies and on the Symptom Check List and its subdimensions [19,20,21]. 

### 2.2. Statistical Analysis

Data were collected in a spreadsheet and analyzed using Prism (v.9.1.0; GraphPad software, La Jolla, CA, USA). Three subjects were not included in the gender-based stratification analysis because they declared “other” for sex. Unpaired (Mann–Whitney test) two-tailed *t*-test for two-group comparison and one-way analysis of variance (ANOVA) using Kruskal–Wallis test for three-group comparison were performed. Chi-square test was employed for comparison of categorical variables A multivariate logistic regression analysis was performed using as demographic variables employment status (professors/researchers, administrative officers, students) and age (20–40 years old, 40–60 years old, over 60 years old), and as COVID-19-related variables: perceptions (extremely worried, moderately worried, not worried), emotions, actions for reducing COVID-19 spread, variations in relationships (none, worsening, improvement), increased stress (no/yes), and privacy (reduced, no variations). A *p* < 0.05 was considered statistically significant.

## 3. Results

### 3.1. Gender Differences in COVID-19 Perception

Most of the participants (96.7%) were worried about the COVID-19 pandemic, while only 3.3% of participants declared that they did not feel COVID-19 was a risk. In particular, 42% of participants were extremely worried (males, 42.3%; and females, 57.7%), while 54.6% were moderately worried (males, 53.7%; and females, 46.2%) (Table 2). Of the remaining participants who were not worried about COVID-19 (*n* = 27), 59% of them were males and 41% females. The most frequent feelings among all participants were exasperation, insecurity, and tranquility. After stratification by gender, females declared more frequently to feel insecure (*p* = 0.0160), to experience insomnia (*p* = 0.0033), sadness (*p* = 0.0007), resignation (*p* = 0.0051), loneliness (*p* = 0.0114), fear (*p* = 0.0004), or tranquility (*p* < 0.0001) compared to males (Table 2). Participants were then divided into three groups based on COVID-19 fear (extremely, moderately, or not worried), and the results were stratified by gender to identify a different distribution of feelings between males and females based on COVID-19 perception. Extremely worried females significantly felt more fearful than males (*p* = 0.0463), while no differences between gender were described for the other feelings, as both females and males felt more frequently insecure, exasperated, sad, afraid, or experienced insomnia (Table 2). In moderately worried participants, females felt more often resigned (*p* = 0.0360), alone (*p* = 0.0245), sad (*p* = 0.0035), and experienced more frequently insomnia (*p* = 0.0174) compared to males, while males felt more often quiet (*p*= 0.0025) compared to females (Table 2). No differences were described for the remaining moods between sexes. In participants who did not see COVID-19 as a danger, females felt more frequently resigned (*p* = 0.0188) compared to males, while no differences were described for the remaining feelings (data not shown). 

### 3.2. Gender Differences in Actions for Containing COVID-19 Spread and in Relationships with Co-Habitants

Participants indicated which action(s) they used for containing COVID-19 spread. Available answers were: (i) going outside only if necessary; (ii) staying at home; (iii) washing hands, wearing a mask, and physical distancing; (iv) surface sanitizing. Overall, females were more likely to sanitize surfaces compared to males (*p* < 0.0001), while no differences were observed for going outside only if necessary (*p* = 0.5759), staying at home (*p* = 0.4916), or washing hands, wearing a mask, and physical distancing (*p* = 0.1702) (Table 3). 

When divided by sex and grade of concern about COVID-19, extremely worried females were more likely to sanitize surfaces compared to males (*p* = 0.0007), while no differences were described for the remaining actions also in moderately and not worried participants (Table 3).

Next, subjects were asked in which ways they have modified their relationships with co-habitants, either a family member or a roommate. Overall, no differences were described between genders in modifications of relationships with co-habitants (Table 4). In particular, 5.3% of females and 4.3% of males declared to live alone, and 42% and 41.4% of females and 43.5% and 43.4% of males answered an intensification in conversation with cohabitants and in doing activities together, respectively. In total, 15.5% of females and a 14.8% of males felt a reduction in privacy. When divided by sex and grade of concern about COVID-19, only moderately worried females experienced significantly increased stress compared to males (*p* = 0.0365). No other gender-related differences were described (Table 4). Among subjects who were not worried about COVID-19 (*n* = 27), only six of them felt an increase in stress (22%; M/F, 5/1), four a lower privacy (15%; M/F, 3/1), and 15 intensified relationships (55%; M/F, 8/7). 

A multivariate logistic regression analysis was performed to investigate the influence of gender on employment status, age, COVID-19 perception, feelings, actions for reducing COVID-19 spread, variations in relationships, increased stress, and privacy (Table 5). Age (20–40 years old, *p* = 0.0123), resignation (*p* = 0.0051), and tranquility (*p* = 0.0295), and surface sanitation (*p* = 0.0111) were related with gender, while no other associations were described. 

### 3.3. Gender Differences in Teleworking Experience during COVID-19 Pandemic

During the first wave of the pandemic, 83.2% of participants teleworked all working days, while 11.9% of subjects alternated teleworking and working on place, and only 4.9% of people continued to work on place. Among subjects who teleworked, 36% of people declared to have a satisfactory number of electronic devices for themselves and their family needs. Moreover, 36.9% of participants had an adequate internet connection for teleworking, while 53.8% of subjects declared that internet quality was just sufficient for their needs. Only 13.7% and 9.2% of participants did not have an adequate number of devices or a good internet connection, respectively. Overall, no differences between females and males were described in work performance change (*p* = 0.3294), and 83.4% or 88% of females or males with children had a positive experience with teleworking during the pandemic (*p* = 0.6953) (Figure 1A,B). Moreover, 67% of participants had an improvement in their private life in both females and males (*p* = 0.4603) (Figure 1C). 

Next, the impact of home-to-workplace distance and the number of family members on positive experience with teleworking was investigated. After stratification by gender, no impact of the number of family members or home-to-workplace distance on positive teleworking experience was described between males and females (*p* = 0.3532 and *p* = 0.8065, respectively) (Figure 1D,E). Finally, more than 51% of participants were favorable to continue teleworking after the pandemic, especially partial teleworking (few days/week), or only if necessary, especially females (*p* = 0.0357) (Figure 1F).

## 4. Discussion

The novel coronavirus SARS-CoV-2 is responsible for the COVID-19 pandemic characterized by an acute respiratory distress syndrome [22]. Infected people might show a longer and complex disease also involving cognitive functions (the so-called brain fog) and psychiatric symptoms [23], while non-infected people have experienced psychological sequalae related to long-term confinement due to shelter-at-home mandates and to the fear of developing severe COVID-19 [16,17,22,23,24]. In this study, we highlighted gender differences in COVID-19 perception in a university setting showing that females were more vulnerable to negative impacts of the pandemic compared to males. We also highlighted a difference in increased attention to surface sanitation to reduce COVID-19 risk of infection. 

Mental health sequelae are related to a direct experience of COVID-19 symptoms without testing, a positive test result, and a family member or a close friend dying from the infection [13,25,26,27]; however, psychological problems after the pandemic can be also linked to quarantine or stay-at-home mandates lasting for a long period of time [27]. Moreover, job losses, unemployment-related stress, and economic difficulties can considerably enhance the psychosocial burden of COVID-19 outbreak [28]. Several studies have already reported that females of all ages and education grades are more vulnerable than males, likely because they frequently need to take care of children and family [13]. In this study, we showed that females with high-grade education are more exposed than males to the psychological burden of a pandemic and perceived the novel coronavirus infection as a very dangerous health problem. Indeed, in our cohort, females were more likely to be extremely worried and afraid than males about COVID-19. In addition, moderately worried females felt more resigned, alone, and sad than males and more frequently suffered from insomnia; conversely, males were more serene than females even though they perceived COVID-19 as an extremely or moderately serious problem. These sex differences in coping strategies and crisis perceptions have already been described, and females are more likely to use a less positive reframing that subsequently leads to an increased incidence of depression and anxiety [29,30,31,32]. Indeed, females reported an increased frequency of sleep disorders during the first wave, which could be a manifestation of mood disorders, as already described [33]. Conversely, males are predisposed to more easily and quickly accepting and overcoming negative situations, partially explaining why a subgroup of male participants of our survey indicated feeling calmer than females during the first lockdown and paying less attention to surface sanitation. A survey from eight countries (Australia, Austria, France, Germany, Italy, New Zealand, the United Kingdom, and the United States) has shown large gender differences in COVID-19 perception as a health problem during the first and second wave [34]. This survey remarks that results are generalizable because countries with different COVID-19 mortality rates (e.g., the United Kingdom, the United States, and Italy) show common gender differences in response to the pandemic [34,35,36]. According to these observations, we described a more serious individual perception of COVID-19 as a health problem in females in the south of Italy, even though the south was not affected as much as the north of Italy during the first wave of the pandemic (on 1 June 2020, total number of cases in Lombardy = 89,018 with a 18.1% mortality vs. total number of cases in Campania = 4806 with an 8.6% mortality). Therefore, our data confirmed a gender difference in the perception of the seriousness of COVID-19 regardless of mortality rate, grade of education and job positions [36,37,38].

This higher awareness of females and fear of the COVID-19 contagion could be linked to a greater willingness of women to obey restraining public policy measures, such as closing schools and nonessential shops or institutions, prohibiting nonessential travel and gatherings, quarantine on people entering the country, physical six-foot distancing, wearing masks indoors and outdoors, or wearing gloves [34]. In our study, we showed no gender differences in adherence to restrictions for containing COVID-19 spread during the first wave of the pandemic, possibly because our respondents had medium or high-grade education and worked in an academic setting that supported teleworking. Only surface sanitation was significantly more frequently performed by extremely and moderately worried females, likely because of an underlying increased fear of COVID-19 infection. 

Finally, there is contrasting evidence of teleworking experience during the COVID-19 pandemic, as several studies are also showing negative long-term impacts of teleworking on daily life, physical and mental health, and relationships with cohabitants [39,40,41,42]. An Italian survey conducted during the first wave in the north of Italy has highlighted an increased cognitive demand and off-work hour technology-assisted job demands, especially for women with high responsibilities [40]. In addition, teleworking has been linked to increased information and communications technology-related stress, food intake, distractions while working, and less physical exercise [43,44,45]; however, some studies and surveys report a positive experience with teleworking because of a reduction in unnecessary travel and in stress related to traveling every day to work, especially long distances or in crowded places, and even in finding parking [46]. In our analysis, the majority of subjects had a positive experience with teleworking, especially females with children and males with a large family (more than two children) or travelling every day for more than 50 km, and more than 60% of our subjects felt an improvement in family life with teleworking and wished to maintain this work approach even after the pandemic, especially administrative officers. Finally, females declared to be favorable to maintain teleworking after COVID-19 emergency only if institutions could offer appropriate childcare services and allow an individual modulation of workhours/week (data not shown). The main limitation of our work is that this study was performed on a small sample size and enrolling people with a medium-to-high degree of education in an academic setting; therefore, the results were not representative of the entire population of Salerno city and county. 

## 5. Conclusions

In conclusions, our single-center study conducted at the University of Salerno, in the south of Italy, during the first wave of the COVID-19 pandemic confirmed gender differences in perceptions of new risk, females being more aware than males in facing a new health problem and adhering to restriction measures, while being more susceptible to mental health sequalae likely because of a more negative ability to cope during a crisis period, ultimately leading to an increased incidence of sleep disorders, anxiety and depression. Therefore, females were more compliant and less likely to spread the virus, as they felt more of the psychological burden of the emergency and because they are frequently caregivers and the center of the family; in contrast, males were less likely to strictly obey restriction measures, and they were more prone to positively reframing the pandemic condition, quickly coming back to “normal life”. Moreover, teleworking could represent, for both sexes, an important work strategy to reduce stress related to daily home-to-workplace and unnecessary travels, and to increase the time for activities with family. 

## Figures and Tables

**Figure 1 jpm-12-00613-f001:**
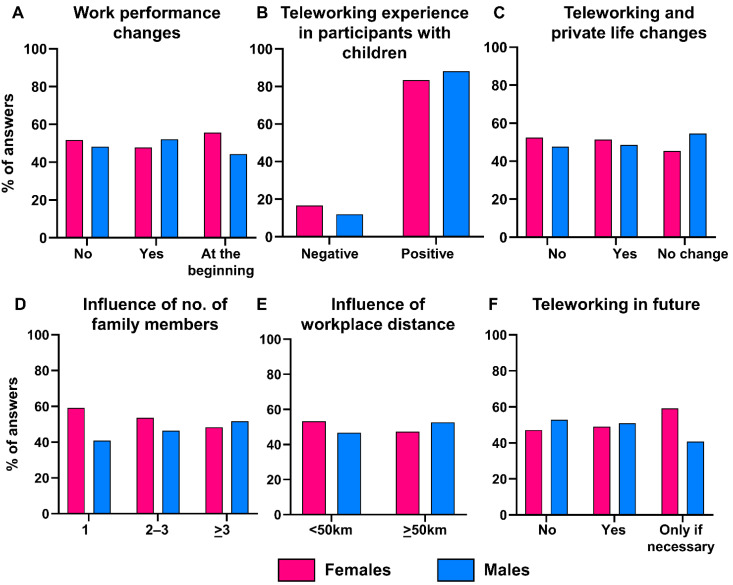
Gender differences and teleworking during COVID-19 pandemic. (**A**) Differences in work performance were stratified by gender and reported as bar graphs. (**B**) Teleworking experience in females with children, and (**C**) influence in private life stratified by gender. Among participants with a positive experience with teleworking, the impact of the number of family members (**D**) and home-to-workplace distance (**E**) was investigated after stratification by gender. (**F**) Percent of answers about willingness to continuing teleworking even after pandemic, stratified by gender.

**Table 1 jpm-12-00613-t001:** Socio-demographic characteristics (*n* = 899).

	Total *n* = 899		Females *n* = 456		Males *n* = 440		*p* Value
	*n*	%	*n*	%	*n*	%	
**Age**							0.0005 *
20–40 years old	276	30.7	163	35.8	112	25.5	
40–60 years old	501	55.7	246	53.9	254	57.7	
>60 years old	122	13.6	47	10.3	74	16.8	
**Employment status**							0.0008 *
Professors/researchers	329	36.6	145	31.8	183	41.5	
Administrative officers	323	35.9	163	35.7	160	36.3	
Students	247	27.5	148	32.5	98	22.2	
**Employment status and age**							
**Professors/researchers**							0.0114 *
20–40 years old	44	13.4	24	16.6	20	11	
40–60 years old	226	68.7	105	72.4	120	65.9	
>60 years old	59	17.9	16	11	42	23.1	
**Administrative officers**							0.2539
20–40 years old	31	9.6	20	12.3	11	6.9	
40–60 years old	230	71.2	112	68.7	118	73.7	
>60 years old	62	19.2	31	19	31	19.4	
**Students**							0.3885
20–40 years old	201	81.4	119	80.4	81	82.7	
40–60 years old	45	18.2	29	19.6	16	16.3	
>60 years old	1	0.4	0	0	1	1	
**Living conditions**							0.6419
Alone	82	9.1	40	8.8	41	9.3	
With 1 person	169	18.8	93	20.4	76	17.3	
With 2 people	236	26.3	121	26.5	115	26.1	
With >3 people	412	45.8	202	44.3	208	47.3	
**Number of children**							0.0382 *
None	366	40.7	201	44.1	164	37.3	
>1	533	59.3	255	55.9	276	62.7	

*, *p* < 0.05.

**Table 2 jpm-12-00613-t002:** Gender differences in COVID-19 perception among participants based on grade of concern.

Emotions	Overall			Extremely Worried			Moderately Worried		
	Females *n* = 456	Males *n* = 440	*p* Value	Females *n* = 218	Males *n* = 160	*p* Value	Females *n* = 227	Males *n* = 264	*p* Value
Resignation	120 (26%)	81 (17%)	0.0051 *	49 (23%)	26 (16%)	0.1518	67 (29%)	55 (21%)	0.0360 *
Insomnia	133 (29%)	90 (20%)	0.0033 *	70 (32%)	40 (25%)	0.1382	62 (27%)	48 (18%)	0.0174 *
Loneliness	76 (17%)	47 (11%)	0.0114 *	35 (16%)	17 (11%)	0.1733	40 (18%)	27 (10%)	0.0245 *
Sadness	129 (28%)	82 (19%)	0.0007 *	72 (33%)	46 (29%)	0.4318	55 (24%)	36 (14%)	0.0035 *
Exasperation	147 (32%)	136 (31%)	0.7195	66 (30%)	43 (27%)	0.4924	78 (34%)	88 (33%)	0.8489
Fear	98 (21%)	55 (13%)	0.0004 *	80 (37%)	43 (27%)	0.0463*	17 (8%)	11 (4%)	0.1233
Insecurity	157 (34%)	117 (27%)	0.0160 *	96 (44%)	66 (41%)	0.6006	56 (25%)	50 (19%)	0.1528
Anxiety	12 (3%)	5 (1%)	0.1409	10 (5%)	2 (1%)	0.0798	1 (0.4%)	3 (1%)	0.6275
Concern	9 (2%)	7 (2%)	0.8024	6 (3%)	3 (2%)	0.7388	3 (1%)	4 (2%)	>0.9999
Tranquility	102 (22%)	151 (34%)	<0.0001 *	35 (16%)	33 (21%)	0.2793	65 (29%)	110 (42%)	0.0025 *
Ease	52 (11%)	53 (12%)	0.7567	20 (9%)	14 (9%)	>0.9999	30 (13%)	38 (14%)	0.7935
Pleasure	23 (5%)	31 (7%)	0.2102	6 (3%)	9 (6%)	0.1869	16 (7%)	19 (7%)	>0.9999

*, *p* < 0.05.

**Table 3 jpm-12-00613-t003:** Gender differences in actions for containing COVID-19 spreading.

Actions	Overall			Extremely Worried			Moderately Worried		
	Females *n* = 456	Males *n* = 440	*p* Value	Females *n* = 218	Males *n* = 160	*p* Value	Females *n* = 227	Males *n* = 264	*p* Value
Go outside only if necessary	385 (84%)	377 (86%)	0.5759	187 (86%)	141 (88%)	0.5420	193 (85%)	226 (86%)	0.8985
Stay at home	89 (20%)	77 (18%)	0.4916	53 (24%)	43 (27%)	0.6327	35 (15%)	33 (13%)	0.3620
Mask/hand sanitation	109 (24%)	123 (28%)	0.1702	38 (17%)	36 (23%)	0.2390	65 (29%)	82 (31%)	0.6213
Surface sanitation	175 (38%)	112 (25%)	<0.0001 *	107 (49%)	50 (31%)	0.0007 *	68 (30%)	59 (22%)	0.0629

*, *p* < 0.05.

**Table 4 jpm-12-00613-t004:** Gender differences in relationships during COVID-19.

Actions	Overall			Extremely Worried			Moderately Worried		
	Females *n* = 456	Males *n* = 440	*p* Value	Females *n* = 218	Males *n* = 160	*p* Value	Females *n* = 227	Males *n* = 264	*p* Value
Intensified	294 (64%)	281 (64%)	0.9820	144 (66%)	110 (69%)	0.3501	143 (63%)	163 (62%)	0.7074
Reduced	49 (11%)	48 (11%)		21 (10%)	20 (13%)		27 (12%)	34 (13%)	
No variations	113 (25%)	111 (25%)		53 (24%)	30 (19%)		57 (25%)	77 (29%)	
Increased stress	70 (15%)	64 (15%)	0.4287	26 (12%)	26 (16%)	0.2279	44 (19%)	33 (13%)	0.0365 *
Lower privacy	71 (16%)	65 (15%)	0.2418	29 (13%)	26 (16%)	0.8029	40 (18%)	36 (14%)	0.2236

*, *p* < 0.05.

**Table 5 jpm-12-00613-t005:** Multivariate analysis.

	OR	95% CI	|Z|	*p* Value
Intercept	1.61	0.6718 to 3.887	1.06	0.29
**Employment status**				
Administrative officers	0.88	0.6325 to 1.227	0.75	0.45
Students	0.76	0.4858 to 1.202	1.17	0.24
**Age**				
20–40 years old	0.49	0.2771 to 0.8536	2.50	0.01
40–60 years old	0.67	0.4345 to 1.027	1.83	0.07
**COVID-19 perception**				
Extremely worried	0.76	0.5497 to 1.040	1.72	0.09
Not worried	0.93	0.3961 to 2.242	0.16	0.87
**Emotions**				
Resignation [NO]	1.62	1.158 to 2.283	2.80	0.01 *
Insomnia [YES]	0.77	0.5495 to 1.074	1.54	0.12
Loneliness [YES]	0.73	0.4679 to 1.128	1.41	0.16
Sadness [YES]	0.76	0.5360 to 1.079	1.53	0.13
Exasperation [YES]	1.04	0.7531 to 1.446	0.26	0.80
Fear [YES]	0.79	0.5228 to 1.202	1.09	0.28
Insecurity [YES]	0.97	0.6973 to 1.342	0.20	0.84
Anxiety [YES]	0.61	0.1816 to 1.793	0.87	0.39
Concern [YES]	0.85	0.2868 to 2.452	0.30	0.77
Tranquility [YES]	1.48	1.040 to 2.104	2.18	0.03 *
Ease [YES]	0.83	0.5171 to 1.331	0.77	0.44
Pleasure [YES]	1.19	0.6467 to 2.231	0.56	0.57
**Actions**				
Go outside only if necessary [YES]	0.72	0.3331 to 1.528	0.86	0.39
Wear mask/wash hands/physical distancing [YES]	1.40	0.8470 to 2.327	1.30	0.20
Surface sanitation [YES]	0.67	0.4951 to 0.9128	2.54	0.01 *
No actions [YES]	3.23	0.3994 to 69.45	0.98	0.33
**Variations in relationships**				
None	1.03	0.7309 to 1.442	0.15	0.88
Worsening	1.27	0.7258 to 2.226	0.84	0.40
**Increased stress [NO]**	1.01	0.6268 to 1.630	0.04	0.97
**Privacy [REDUCED]**	1.16	0.7544 to 1.791	0.68	0.50

Abbreviations: OR: odd ratio; CI: confidential interval. *, *p* < 0.05.

## Data Availability

Data are available upon request by the authors.

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
