# Peer review of "Gender Differences in the Impact of COVID-19 Pandemic on Mental Health of Italian Academic Workers"

_jpm, 2022, doi:10.3390/jpm12040613_

Round 1

Reviewer 1 Report

Dear authors,

I would like to congratulate for very valuable research results, you have been prepared.

However, the Introduction part should be extended. The backbone of your article is the results of empirical research, but in the paper written in this way, there is an obvious disproportion between the introductory and empirical part.

Additionally, I do not understand is why did you put link of your survey in the manuscript, if it does not work? Please see:

https://docs.google.com/forms/d/e/1FAIpQLSdGd6sJRtzFesSsck-iJBGkr-7RWiefbE6-kRI%2063dR77tdq7VNA/closedform

Author Response

Reviewer 1

Dear authors,

I would like to congratulate for very valuable research results, you have been prepared.

Comment 1. However, the Introduction part should be extended. The backbone of your article is the results of empirical research, but in the paper written in this way, there is an obvious disproportion between the introductory and empirical part.

Response to Comment 1. We thank the Reviewer for positive feedbacks and for these suggestions. We have extended the introduction section as follows.

On pages 1-2, lines 41-54, the following text was added “Sex significantly and variously impacts on Covid-19 biology and clinical out-comes, and differences are attributable to sex steroid action or other gender-related physiological differences, such as in type of immune responses [11]. Moreover, variations in females are not only related to sex and age, but also to pregnancy, pre- or post-menopausal conditions. For example, decreased sex hormone levels are related to severe symptoms and increased mortality rate, as described in older males with low free and total testosterone, or postmenopausal females with low 17-β estradiol [11]. Conversely, immune system assets in females, while favoring a better response against viral agents, can cause an exaggerated response after vaccination thus leading to increased frequencies of vaccine-related adverse events and low efficacy [11]. However, sex differences on Covid-19 impact are not only related to a different gender-related biology, but also to environment, behavior, type of job, and habits that could also im-pact Covid-19 incidence and development of psychological disturbances, such as anxiety, distress, depression, and post-traumatic stress disorder (PSTD) [12-14].”

Comment 2. Additionally, I do not understand is why did you put link of your survey in the manuscript, if it does not work? Please see:

https://docs.google.com/forms/d/e/1FAIpQLSdGd6sJRtzFesSsck-iJBGkr-7RWiefbE6-kRI%2063dR77tdq7VNA/closedform

Response to Comment 2. We apologize for this inconvenience. The link was referring to the website where the online survey was available; however, that link has been discontinued by the University thus we have removed it from the text.

Reviewer 2 Report

Major changes
The link for supporting information (www.mdpi.com/xxx/s1, Supplementary Material S1: Survey on Covid-19) is not working.

The online survey link https://docs.google.com/forms/d/e/1FAIpQLSdGd6sJRtzFesSsck-iJBGkr-7RWiefbE6-kRI63 dR77tdq7VNA/closedform is not all the time available. You mention that the questionnaire is available in Italian and English, please include the English version.

Covid-19 related variable perceptions (extremely worried, moderately worried, no worried) is a survey item or is defined by some criteria by you? How many cases were in the category “no worried”? The same question for stress / increased stress. 

In the context of multivariate logistic regression analysis, it is not clear for me which were the predictor variables and which the target?

Minor changes:

Some references are mentioned without [] (ex. line 220, 249)

Author Response

Reviewer #2

Major changes

Comment 1. The link for supporting information (www.mdpi.com/xxx/s1, Supplementary Material S1: Survey on Covid-19) is not working.

Response to Comment 1. We apologize for this inconvenience. This is a default link when submitting a manuscript. Please find below the Supplementary Material S1: Survey on Covid-19, that is the questionnaire used in English version.

Comment 2. The online survey link

https://docs.google.com/forms/d/e/1FAIpQLSdGd6sJRtzFesSsck-iJBGkr-7RWiefbE6-kRI63 dR77tdq7VNA/closedform is not all the time available.

Response to Comment 2. We apologize for this inconvenience. The link was referring to the website where the online survey was available; however, that link has been discontinued by the University thus we have removed it from the text.

Comment 3. You mention that the questionnaire is available in Italian and English, please include the English version.

Response to Comment 3. Please refer to Response to Comment 1.

Comment 4. Covid-19 related variable perceptions (extremely worried, moderately worried, no worried) is a survey item or is defined by some criteria by you?

Response to Comment 4. We thank the Reviewer for this comment. Covid-19 related variable perceptions were survey items and were referring to a subjective feeling of how Covid-19 was perceived as serious event.

Comment 5. How many cases were in the category “no worried”? The same question for stress / increased stress.

Response to Comment 5. We thank the Reviewer for this comment, and we have added missing information.

Lines 106-108, “Of the remaining participants who were not worried about Covid-19 (N = 29), 59% of them were males and 41% females.”

Lines 148-150, “Among subjects who were not worried about Covid-19 (N = 29), only six of them felt an increase in stress (21%; M/F, 5/1), four a lower privacy (14%; M/F, 3/1), and 15 intensified relationships (52%; M/F, 8/7).”

Comment 6. In the context of multivariate logistic regression analysis, it is not clear for me which were the predictor variables and which the target?

Response to Comment 6. Gender (female) was the predictor, and institutional roles, age, Covid worry, resignation, insomnia, loneliness, sadness, exasperation, fear, insecurity, anxiety, concern, tranquility, ease, pleasure, go outside only if necessary, wear mask/wash hands/physical distancing, surface sanitation, no actions, variations in relationships, increased stress, and lower privacy were the targets.

Minor changes:

Some references are mentioned without [] (ex. line 220, 249)

Changed.

Round 2

Reviewer 2 Report

Minor changes:

In Tabel 1:

  • there are 456 females while in Table 2 there are 457 females

  • 456 f + 440 m = 896 cases , while the total is 899

In Table 2: 

  • there are 218+227=445 (total 457-445= 12 females) and 160+267=427 (total 440-427=13 males). The 12+13=25 cases are from the category “no worried”? (on line 119 it is mentioned that there are 29 cases in this category)

In Table 3 and Table 4: 

  • there are 264 males in “moderately worried” category while in Table 2 there were 267 

If there were missing values for some items you should mention (maybe in the material and methods section and/or below the tables).

If I understood correctly, in the case of multivariate regression it was considered (among others) as the target variable age and as the predictor variable gender. How you interpret the influence of gender on age?

Author Response

Minor changes:

Comment 1. In Table 1: there are 456 females while in Table 2 there are 457 females

456 f + 440 m = 896 cases, while the total is 899

Response to Comment 1. We apologize for this typo. We have corrected 456 in the other tables.

Comment 2. In Table 2: there are 218+227=445 (total 457-445= 12 females) and 160+267=427 (total 440-427=13 males). The 12+13=25 cases are from the category “no worried”? (on line 119 it is mentioned that there are 29 cases in this category)

Response to Comment 2. We apologize for this. Total number of females is 456, therefore in the category “no worried” there are 11 females; while for males, there are 160 extremely+264 moderately worried, therefore there are 16 males in the no worried category. Total number of no worried subjects is 27. We have changed the text according as follows.

On lines 148-150, the text was modified as follows “Among subjects who were not worried about Covid-19 (N = 27), only six of them felt an increase in stress (22%; M/F, 5/1), four a lower privacy (15%; M/F, 3/1), and 15 intensified relationships (55%; M/F, 8/7).”

Comment 3. In Table 3 and Table 4: there are 264 males in “moderately worried” category while in Table 2 there were 267.

Response to Comment 3. We apologize for the typo, now it reads 264 as in the other tables.

Comment 4. If there were missing values for some items you should mention (maybe in the material and methods section and/or below the tables).

Response to Comment 4. In the 2.2 Statistical analysis section, lines 89-90, the following text was added “Three subjects were not included in the gender-based stratification analysis because they declared “other” for sex.”

Comment 5. If I understood correctly, in the case of multivariate regression it was considered (among others) as the target variable age and as the predictor variable gender. How you interpret the influence of gender on age?

Response to Comment 5. We thank the reviewer for this comment. We might explain this result with the fact that females are more represented in certain jobs compared to males, especially in recent years with the promotion of gender equality. As yearly reported by the Global Gender Gap Report (http://www3.weforum.org/docs/WEF_GGGR_2020.pdf), females tend to work more in “people and culture”, “content production”, and “marketing” categories rather than engineering. This trend is growing in recent years for these categories, while slowly increasing for others, such as political and economic participation. We might suggest that the influence of gender on age in our cohort might reflect this recent tendency of females to prefer to work in science rather than in other fields.